# Comprehensive Study of Antiretroviral Drug Permeability at the Cervicovaginal Mucosa *via* an In Vitro Model

**DOI:** 10.3390/pharmaceutics14091938

**Published:** 2022-09-13

**Authors:** Constandinos Carserides, Kieron Smith, Marta Zinicola, Abhinav Kumar, Magda Swedrowska, Carlo Scala, Gary Cameron, Zoe Riches, Francesco Iannelli, Gianni Pozzi, Georgina L. Hold, Ben Forbes, Charles Kelly, Karolin Hijazi

**Affiliations:** 1Centre for Host Microbiome Interactions, King’s College London, London SE1 9NH, UK; 2Institute of Pharmaceutical Science, King’s College London, London SE1 9NH, UK; 3School of Medicine Medical Sciences & Nutrition, University of Aberdeen, Aberdeen AB25 2ZR, UK; 4Laboratory of Molecular Microbiology and Biotechnology, Department of Medical Biotechnologies, University of Siena, 53100 Siena, Italy

**Keywords:** HIV-1, female genital tract, darunavir, tenofovir, dapivirine, drug permeability, drug transporters, dual chamber model, HEC-1A

## Abstract

Modulation of drug transporter activity at mucosal sites of HIV-1 transmission may be exploited to optimize retention of therapeutic antiretroviral drug concentrations at target submucosal CD4+ T cells. Previously, we showed that darunavir was a substrate for the P-glycoprotein efflux drug transporter in colorectal mucosa. Equivalent studies in the cervicovaginal epithelium have not been reported. Here, we describe the development of a physiologically relevant model to investigate the permeability of antiretroviral drugs across the vaginal epithelium. Barrier properties of the HEC-1A human endometrial epithelial cell line were determined, in a dual chamber model, by measurement of transepithelial electrical resistance, immunofluorescent staining of tight junctions and bi-directional paracellular permeability of mannitol. We then applied this model to investigate the permeability of tenofovir, darunavir and dapivirine. Efflux ratios indicated that the permeability of each drug was transporter-independent in this model. Reduction of pH to physiological levels in the apical compartment increased absorptive transfer of darunavir, an effect that was reversed by inhibition of MRP efflux transport *via* MK571. Thus, low pH may increase the transfer of darunavir across the epithelial barrier *via* increased MRP transporter activity. In a previous in vivo study in the macaque model, we demonstrated increased MRP2 expression following intravaginal stimulation with darunavir which may further increase drug uptake. Stimulation with inflammatory modulators had no effect on drug permeability across HEC-1A barrier epithelium but, in the VK2/E6E7 vaginal cell line, increased expression of both efflux and uptake drug transporters which may influence darunavir disposition.

## 1. Introduction

Intra-vaginal application of antiretrovirals (ARVs) as pre-exposure prophylaxis in forms such as gels, films and vaginal rings allows for greater accumulation of these drugs within the cervicovaginal mucosa, an environment rich in CD4+ cells targeted by the human immunodeficiency virus (HIV)-1. For optimal protection against HIV-1 transmission, ARV accumulation at therapeutic levels must be achieved and sustained at the submucosal CD4+ target cells [1]. Tight junction formation in the columnar epithelial layer of the endocervix and the basal layers of the stratified squamous epithelium of the ectocervix and vagina may limit paracellular transport of ARVs into the cervicovaginal submucosa. In addition, hydrophobic ARVs may partition into epithelial cell membranes, further limiting effective concentrations of ARVs at HIV-1 target cells.

In vitro and in vivo studies demonstrated that some ARVs, under development as vaginal microbicides, can modulate the expression of drug transporters in cervicovaginal cells [2,3] and are substrates for uptake and efflux drug transporters [4], many of which are present in cervicovaginal cells [2,5,6,7]. Thus, transfer across the cervicovaginal epithelial barrier may be drug transporter-dependent and modulation of drug transporter activity may be exploited to maximise retention of therapeutic ARV concentrations at target cells.

We have previously investigated the permeability of ARVs in colorectal mucosa and found that darunavir showed P-glycoprotein (P-gp)-dependent permeability while uptake of tenofovir was predominantly paracellular [8]. Equivalent studies of cervicovaginal mucosa investigating ARV permeability and drug transporter-dependence across the cervicovaginal mucosa are limited to tenofovir and describe permeability in a dual-chamber model of HEC-1A monolayers grown at neutral pH [9]. However, the effect of environmental factors such as acidic pH, microbial dysbiosis, inflammation and hormones, all of which may impact the expression of ARV-interacting drug transporters, should be considered in the formulation of vaginally applied drugs.

The modulatory role of pH on drug transporter expression and function has been investigated in other models. The functional activity of P-gp in prostate carcinoma cells more than doubled after 3 h of incubation at pH 6.6 versus control pH 7.4 [10]. An increase in the functional activity of OATPB in transfected HEK293 cells was evident at pH 5.5 compared with pH 7.4 [11].

The acidic pH of the cervicovaginal tract is maintained by lactic acid production of commensal bacteria such as Lactobacillus spp., which may also play a protective role by competitively preventing mucosal colonisation by species associated with vaginosis and vaginitis such as *Gardnerella vaginalis* and *Candida albicans* [12]. These microbial species may affect drug transporter expression directly or as a result of microbially-induced inflammation. Concentrations of pro-inflammatory cytokines in cervicovaginal secretions vary depending on the relative proportions of Lactobacillus spp. versus potentially pathogenic species [13,14,15].

Modulation of drug transporter expression by specific cytokines has been demonstrated in various cell types, but not in cells derived from cervicovaginal tissue. Expression and activity of P-gp, MRP1 and MRP2 in peripheral mononuclear blood cells increased in response to stimulation with Th1 cytokines (IL-2, IL-12, IFNγ), Th2 cytokines (IL-4, IL-6, IL-10, IL-13) and, at a lower level, non-Th1/Th2 cytokines (IL-7, IL-15, TGF-β, TNF-α) [16]. In contrast, TNF-α and IL-1β downregulated the expression of P-gp and BCRP, by 40–50%, in placental trophoblasts [17]. In human hepatocyte models, IL-1β, IL-6 and TNF-α also downregulated the expression of several SLC uptake transporters (including OATPB, OATPC, OATP8, OCT1 and OCT2) and ABC efflux transporters but upregulated MRP3 [18,19].

Female sex hormones β-estradiol and progesterone, the levels of which vary throughout different stages of the menstrual cycle and between different individuals [20], modulate drug transporter expression. Estradiol upregulated efflux transporters BCRP, MDR1 and MDR3 in primary placental trophoblasts, whilst progesterone upregulated MRP1 in the same model [17]. MDR1 was also upregulated in MDCKII cells following stimulation with β-estradiol or progesterone [20].

In cervicovaginal tissue, the effect of environmental factors on drug permeability and expression of drug transporters themselves has not been evaluated, despite their well-documented effect on drug transporters in other tissues. This gap in knowledge hampers conclusive evidence regarding the role of efflux or uptake drug transporters in ARV permeability through the cervicovaginal epithelial barrier. In this study, we aimed to (i) establish an in vitro model of vaginal barrier epithelium using the human endometrial epithelial cell line HEC-1A, (ii) investigate the uptake of three topically applied ARVs, (iii) investigate the effects of pH and inflammation on ARV uptake and drug transporter-dependent permeability across the vaginal barrier epithelium, (iv) investigate the effect of pH, inflammation, vaginal microbiota and hormones on the expression of drug transporters in the vaginal epithelium.

## 2. Materials and Methods

### 2.1. Cell Lines, Growth Media, Drugs and Cell Stimulants

Cell lines were from American Type Culture Collection (ATCC, Manassas, VA, USA). The HEC-1A cell line (ATCC HTB-112 was grown in McCoy’s 5A Modified Medium supplemented with 10% FBS (medium pH was 7.4 unless otherwise stated). HEC-1A cells were also adapted for culture at reduced pH to assess the effects of pH on drug transport. For these experiments, 1 mM DL-lactic acid was added to the cell culture medium to reduce pH to 5, incrementally by 0.25 pH unit. After each incremental 0.25 pH unit reduction, cells were maintained at the reduced pH for 24 h and subsequently cultured at pH 7.4 for another 24 h before another 0.25 pH unit reduction. A reduction in cell viability was evident at pH 5 (as determined by XTT assays), pH 5.5 was chosen as the lowest limit for assays aimed at testing the effect of acidic pH on drug permeability and drug transporter expression.

The VK2/E6E7 (ATCC CRL-2616) cell line, derived from primary vaginal epithelial cells, was grown in Keratinocyte-Serum Free Medium (KSFM) supplemented with 0.1 ng/mL human recombinant Epidermal Growth Factor (EGF), 0.05 mg/mL Bovine Pituitary Extract and 0.4 mM CaCl_2_. Culture media were supplemented with 100 U/mL penicillin and 100 μg/mL streptomycin. Cell lines were all grown for a maximum of 5 passages before analyses.

All chemicals and reagents used for the culture of HEC-1A and VK2/E6E7 cells were from Sigma-Aldrich (Poole, UK).

Tenofovir (kindly provided by Johnson and Johnson, New Brunswick, NJ, USA) was solubilised in HBSS. Dapivirine (purchased from Selleckchem, Houston, TX, USA) and Darunavir (kindly provided by Janssen R&D Ireland, Cork, Ireland) were dissolved in DMSO as stock solutions at 100 mM and 250 mM, respectively, and stored at −70 °C. For transport studies, darunavir and dapivirine were diluted in HBSS by incubation at 37 °C for 10 min, with regular vortexing to ensure complete solubilisation of the drug. MK571 (Sigma-Aldrich) stock solution was 10 mM in DMSO and was diluted 1:200 in buffered HBSS to 50 µM in 0.5% DMSO. Digoxin (Sigma-Aldrich) stock solution was 5 mM in DMSO was diluted 1:1000 and also in buffered HBSS to 5 µM in 0.1% DMSO. The purity of the three test drugs was 100% according to the providers while digoxin purity was >96.5% which meets the accepted quantity assurance criteria for HPLC; full product specifications are reported on the manufacturer’s websites.

*Lactobacillus crispatus* M247 was previously isolated from the faeces of a human newborn [21]. *Candida albicans* SC5314 reference strain (http://www.candidagenome.org/Strains.shtml#SC5314; accessed on 1 September 2022) was a kind donation from the Aberdeen Fungal Group, University of Aberdeen. All cytokines were obtained from Life Technologies (Life Technologies, Paisley, UK) and reconstituted as per manufacturer recommendations. Progesterone and β-estradiol were purchased from Life Technologies (Life Technologies) and reconstituted in 100% ethanol as stock solutions of 7.5 mM 1 mM, respectively.

### 2.2. Transwells^®^

The apical chambers of 12-well polycarbonate and polyester systems (Corning Costar Corp, Cambridge, UK) were seeded with different densities of HEC-1A cells in 500 µL of the medium while 1500 µL culture medium was added to the basolateral chamber. -Medium in both chambers was changed every 48–72 h.

For some experiments, prior to seeding with HEC-1A cells, semi-permeable apical chamber membranes were coated with human or murine Laminin (Sigma-Aldrich) or reconstituted basement membrane preparation (Matrigel^®^, Corning Costar Corp). Laminin and Matrigel^®^ solutions were diluted in PBS or ice-cold serum-free cell culture media, respectively at varying concentrations as per the manufacturer’s instructions. Once coated, Transwell^®^ permeable supports were air dried for 16–24 h in a laminar flow hood. Any remaining solution was aspirated from the inserts which were then washed once with either PBS or cell culture medium.

### 2.3. Measurement of Transepithelial Electrical Resistance (TEER)

Transepithelial Electrical Resistance (TEER) of HEC-1A cell monolayers grown on permeable supports was measured with an Epithelial Voltohmmeter (EVOM2, World Precision Instruments, Sarasota, FL, USA). All TEER measurements were performed in a laminar flow hood within approximately 3 mins after the removal of Transwells^®^ from the incubator to reduce temperature-induced fluctuation in TEER values. An alternating current with a square waveform was applied at a frequency of 12.5 Hz to measure resistance (Rsample) across the HEC-1A cell layer supported on the semi-permeable membrane, and resistance (Rblank) across the semi-permeable membrane only (no cells). The cell-specific resistance Rnet (= Rsample − Rblank), is multiplied by the surface area of the permeable support to give the TEER value of the cell layer in ohms(Ω) × cm^2^.

Ca^2+^ dependency of the epithelial cell barrier was investigated by the addition of EDTA (2.5 mM) in the medium to the apical chamber while an EDTA-free cell culture medium was added to the basolateral chamber. Cells were incubated at 37 °C. TEER was monitored at various time points over 90 mins. After this period, HEC-1A cell layers were then washed twice with HBSS and the medium in both chambers was replaced with a fresh EDTA-free cell culture medium. HEC-1A cell layer recovery was allowed to proceed, and TEER was measured at 2.5 h and 24 h.

### 2.4. Immunofluorescence

HEC-1A cells were seeded on 0.69 cm^2^ glass slides (BD Biosciences, Oxford, UK) or Transwells^®^. Cells were washed two times with 100 µL HBSS and fixed with 100 µL of 4% (*v*/*v*) paraformaldehyde (Sigma-Aldrich) in 10× PBS for 15 min at 37 °C in 5% CO_2_. Fixed cells were washed three times in 100 µL HBSS then permeabilised by adding 100 µL Triton X-100 (Sigma-Aldrich), (0.2%, *v*/*v* for 15 min at 37 °C in 5% CO_2_). Triton X-100 was removed by washing two times in HBSS, and non-specific binding sites blocked by adding 100 µL TBS (Tris Buffered Saline, Severn Biotech, Kidderminster, UK) 0.1% Tween 20 (Sigma-Aldrich) supplemented with 3% BSA (Bovine Serum Albumin, Sigma-Aldrich) for 30 min at 37 °C in 5% CO_2_. The blocking solution was removed and cells were then incubated for 90 min with anti-ZO-1 monoclonal primary antibody at 2.5 µg/mL (clone 1, BD Bioscience) diluted in TBS-T/BSA. Omission of the primary antibody and use of an irrelevant isotype-matched antibody (anti-gp41) served as negative controls. Cells were washed three times with 100 µL HBSS and incubated with FITC goat anti-mouse IgG conjugated secondary (Sigma-Aldrich) diluted 1:100 in TBS-T/BSA for 2 h at room temperature in the dark. Slides were washed three times with HBSS before removal of the plastic separating chamber. The membranes of the Transwell inserts were cut out with a scalpel (Sigma-Aldrich) and placed on a glass microscope slide (Sigma-Aldrich). To each membrane, Vectashield mounting medium (Vector Laboratories, Burlingame, CA, USA) containing Dapi was then added and a microscope cover slip (Fisher Scientific, Loughborough, UK) was gently applied. Samples were visualised using an Olympus BX61 microscope that was equipped with an Olympus XM10 digital monochrome camera and images were observed by the Olympus Cell F imaging software.

### 2.5. Drug Permeability Studies

All transport experiments were undertaken over a 2-h period (unless stated) at 37 °C in HBSS supplemented with 1% HEPES buffer (Sigma-Aldrich) at pH 7.4 as reported [22]. HBSS was used to prevent the drug from binding to albumin present in cell culture media. All solutions were warmed to 37 °C and experiments were performed on a heated orbital shaker set at 50 RPM. Two HBSS-based transport solutions were prepared: a donor solution containing drugs generally solubilised in HBSS + 1% HEPES +/− DMSO (0.01–1%) to final drug concentrations of 0.1–100 µM, depending on the drug being investigated, and a receiver solution containing HBSS + 1% HEPES +/− poloxamer 407 (0.2% *w*/*v*), (Sigma-Aldrich). Prior to the initiation of the drug transport experiments, TEER was measured. Cell layers were then washed twice with pre-warmed HBSS+1%HEPES transport buffer and equilibrated in the same solution for 30 min at 37 °C. In all permeability assays, donor solutions were spiked with approximately 4000 Becquerel (Bq) of either radioactive [^14^C]mannitol or [^3^H]mannitol. The experiment was initiated when either the donor or receiver solutions were added to the respective compartments. Following the addition of donor solution, 20 µL samples were taken immediately from donor and receiver compartments, T = 0 (for assessment of initial concentration and background radioactivity, respectively). Transwells^®^ were then placed on an orbital shaker set at 50 RPM. Samples, either 200 µL (apical) or 600 µL (basolateral), were taken from the receiver compartments at time points T = 30, T = 60, T = 90 and T = 120 and added to polyethylene liquid scintillation vials. Samples were taken in triplicate and replaced with equivalent volumes of the receiver solution. After final sampling, TEER was measured to confirm that there was no disruption to the epithelial barrier during the experimental procedure. Radioactive samples were mixed with 4 mL of scintillation fluid (Fisher Scientific, Waltham, MA, USA) and analysed using a liquid scintillation counter (Beckman Coulter, Brea, CA, USA). The semipermeable membranes of the Transwell^®^ inserts were cut and also placed into scintillation vials for mass balance measurements. Apparent permeability (P_app_) was calculated using the following equation: P_app_ (cm/s) = (∆Q/∆t)/(AC0V) where ∆Q/∆t is the gradient of the (linear) slope of the cumulative amount of drug in the receiver given as disintegrations per minute (DPM) vs. time, A is the surface area of the Transwell^®^ membrane, C0 denotes the initial loading concentration in the donor chamber at T = 0 and V is the volume of the initial receiver sample taken at T = 0.

### 2.6. Intracellular Accumulation Studies

VK2/E6E7 cells were seeded at 80–90% confluence before medium removal and rinsing with buffered HBSS (pH 7.4). To each well, 3 mL of buffered HBSS was added and the plates were incubated for 30 min at 37 °C/5% CO_2_. The buffer was then replaced with 3 mL of buffered HBSS containing the test compound (5 µM digoxin or 10 µM darunavir) and returned to the incubator onto an orbital plate shaker at 50 rpm for 1 h. A 500 μL sample from each well was stored and the remainder of the buffer was aspirated from the plates. The wells were then washed twice with ice-cold PBS to stop any further uptake of the drug. To each well, 500 μL of ddH_2_O was added and cells were removed using a cell scraper. The cell suspension was then transferred to individual tubes for mechanical cell lysis using a 21-gauge needle followed by a freeze-thaw cycle at −80 °C. Protein content for normalisation calculations was measured *via* a bicinchoninic acid assay. The solution was then centrifuged at 14,000× *g* for 15 min and the supernatants were stored at −80 °C until analysis by LC-MS/MS.

### 2.7. LC-MS/MS

Samples were centrifuged at 14,800 rpm for 5 min to remove cell debris. Supernatants (40 μL) were mixed with 60 μL stavudine (160 pmol in methanol) as an internal standard for darunavir or digotoxigenin (150 pmol in methanol) as an internal standard for digoxin. Analysis was performed on a Thermo Surveyor TSQ Quantum system using electrospray ionisation in positive ion mode. Samples (5 μL) were applied to a C18 column (Gemini 3μ, 150 × 3.0 mm, Phenomenex, Macclesfield, UK) maintained at 40 °C. Darunavir and stavudine were resolved by gradient elution with a mobile phase consisting of A: water and B: 100% methanol, both containing 0.1% formic acid; 5% B was increased to 95% B over 5 min before returning to initial conditions for 4 min re-equilibration for a total run time of 9 min. Digoxin and digitoxigenin were resolved using gradient elution with a mobile phase consisting of A: 30 mM ammonium formate, pH 3.4 and B: methanol; 70% B was increased to 100% B over 3 min before returning to initial conditions for 3 min re-equilibration for a total run time of 6 min. The flow rate for both methods was 350 μL/min. MS detection was performed in single reaction monitoring (SRM) mode using the following precursor to product ion transitions and collision energies: darunavir, m/z 548.1–392.1 @ 10 V and stavudine, m/z 247.1–149.1 @ 10 V, digoxin, m/z 798.4–97.1 @ 31 V and digitoxigenin, m/z 330.1–158.2 @ 8 V. The retention times of darunavir, stavudine, digoxin and digitoxigenin were 7.60, 5.31, 3.27 and 3.75 min respectively. Calibration standards were prepared in the range 0.02–6 μM (darunavir), or 0.01–3 μM (digoxin).

### 2.8. Stimulation with Pathogen-Associated Molecular Patterns (PAMP) and Cytokine Detection

HEC-1A cells were stimulated with different PAMPs (pam 3CSK4, lipoteichoic acid, poly(I:C), *Escherichia coli* lipopolysaccharide, FSL-1, flagellin, imiquimod, zymosan) (InvivoGen, San Diego, CA, USA) diluted in cell culture media with incubation at 37 °C in 5% CO_2_ for 24 h. PAMPs were titrated to determine the optimal concentration for cytokine stimulation. Blank samples contained media only. After 24 h, the supernatant was collected for cytokine detection. Cytokine levels were determined using a Magnetic Luminex^®^ Performance Assay kit (R&D Systems, Minneapolis, MN, USA) following the manufacturer’s instructions. Analyte-specific antibodies were pre-coated onto colour-coded microparticles and detection was carried out with a biotinylated antibody cocktail specific to the target cytokines and a streptavidin-phycoerythrin conjugate. Samples were resuspended in 100 µL of Washing Buffer and measured for a phycoerythrin-derived signal using a Luminex^®^ 100 instrument.

### 2.9. Cell line Stimulation with Drugs, Cytokines, Hormones and Microbes

Dapivirine and Darunavir were added to cells (4 × 10^5^ cells/well) at 10 μM and 250 μM, respectively, in 0.1% DMSO. Control was 0.1% DMSO with no drug. Cytokines were reconstituted in sterile and distilled water and added individually to cell cultures at the following concentrations: TNF-α (200 pg/mL), IL-1β (8 ng/mL), IL-6 (5 ng/mL) and IL-8 (100 ng/mL) were. For hormone stimulation, 100 nM β-estradiol and 750 nM progesterone in 0.1% ethanol were added individually with 0.1% ethanol as control. Cells were stimulated for either drugs, cytokines or hormones for 72 h, before harvesting by centrifugation at 300× *g* for 10 min at 4 °C. For *Lactobacillus crispatus* a final MoI of 10 was used (10:1, bacterial cells:mammalian cell ratio) for co-culture in both HEC-1A and VK2/E6E7 cell lines. For *Candida albicans*, a final MoI of 1 for HEC-1A cells and 0.25 for VK2/E6E7 cells was used. Cells were incubated with fungal cells at 37 °C/5% CO_2_ for 24 h. All cell line stimulation experiments were conducted as a minimum of three biological repeats.

### 2.10. Extraction of RNA, cDNA Synthesis and Real-Time Quantitative PCR

RNA extraction was performed using the RNeasy^®^ Mini Kit (Qiagen, Manchester, UK). Cells were lysed by suspension in 600 μL of RLT Buffer with 1% β-mercaptoethanol (Sigma-Aldrich) and mechanical lysis into a QIAshredder (Qiagen). After the addition of 70% ethanol, the solution was loaded onto RNeasy columns for RNA purification as per the manufacturer’s instructions. DNase treatment was 15 min at room temperature. RNA was eluted from the columns with 30 μL of RNase-free ddH_2_O. Tapestation analysis was performed using an Agilent 2200 Tapestation system to determine the integrity of the RNA.

Synthesis of cDNA was performed using a Superscript^®^ VILO™ kit (Life Technologies, Paisley, UK). Reverse transcription was performed at 25 °C for 10 min, 42 °C for 60 min and 85 °C for 5 min. The cDNA was then stored at −20 °C until RT-PCR assays.

SYBRgreen^®^ PrecisionTM 2× qPCR Mastermix with ROX quencher (Primer Design, Southampton, UK) was used for all RT-qPCR assays. Individual primers were designed to target the HPRT1 and RPLP0 housekeeper genes (selected from an initial panel based on SD scores as a measure of the consistency of expression amongst all samples) in addition to seventeen ARV-associated drug transporters (9 ABC transporters and 8 SLC transporters) replicating the system contained in the TaqMan^®^ 96-well Drug Transporter Array system (Life Technologies, Paisley, UK). Primer sequences are listed in Table 1). Real-time quantitative PCR was performed using the 7900HT Fast Real-Time PCR system and data was analysed using the SDS 2.4 software (Applied Biosystems, Waltham, MA, USA). Cycling conditions were: 95 °C for 5 min, followed by 40 cycles of denaturation at 95 °C for 10 s, annealing at 60 °C for 10 s and extension at 72 °C for 10 s. Relative transcript levels in drug-stimulated cells were determined using the comparative Ct method (ΔΔCt method) [23]. Fold-change values of ±2–3 were considered minor, ±3–5 as moderate and >±5 as high.

### 2.11. Statistical Analysis

Stimulation data and transport data were presented as the means ± SD after normality testing with the Kolmogorov-Smirnov test. For gene expression of cells stimulated with drugs, cytokines, hormones or microbial cells, levels of mRNA expression between relevant controls (no stimulant) and each stimulation condition were compared using a two-tailed Student’s *t*-test. For transport studies, the experimental condition and controls were compared using one-way ANOVA followed by Bonferroni correction of multiple comparisons. Values of *p* < 0.05 were considered statistically significant.

## 3. Results

### 3.1. Dual Chamber Cell Culture Model of Barrier Epithelium

We compared barrier properties of HEC-1A cultures cultured on polyester or polycarbonate semi-permeable membranes in the dual chamber system. The maximum barrier function of HEC-1A cultures on both membranes was reached between 5 and 7 days of culture and maintained for a further 3–5 days. TEER values were higher in cells grown on polyester compared with polycarbonate membranes (maximum approximately 220 Ω × cm^2^ and 80 Ω × cm^2^, respectively; Appendix A). Polyester membranes were used in subsequent experiments. The optimum seeding cell density was 2 × 10^5^ cells/Transwell^®^ insert (surface area 1.12 cm^2^). At higher density, there was no difference in the maximum TEER values or in the time taken to reach this value while at lower density, TEER values increased more slowly (data not shown).

The effect of extracellular matrix proteins (laminin, Matrigel^®^) on barrier epithelium formation was also investigated in this model. When the semi-permeable membrane was coated with human laminin (5 µg/cm^2^) prior to culture, TEER values were higher initially than those observed when no coating was applied, however, between days 6 and 10 there were no significant differences between the two conditions (Appendix A). Similarly, when the semi-permeable membrane was coated with Matrigel^®^ (15 µg/cm^2^), there was no significant difference in the TEER values observed for coated or uncoated cultures (Appendix A).

To determine the length of time that maximum barrier integrity was maintained, HEC-1A cells were cultured in dual chamber inserts for 21 days. Barrier integrity was assessed by measurement of TEER and by determination of the bi-directional permeability of the para-cellular marker ^3^H-mannitol. TEER values reached a maximum value (approximately 250 Ω × cm^2^) on day 6, decreased gradually to 200 Ω × cm^2^ by day 10 and then more rapidly by day 14. P_app_ values for mannitol reached a minimum value of 1.5 × 10^−6^ cm/s on day 6 and remained low until day 10 (Appendix A). Maintenance of the epithelial barrier was Ca^2+^-dependent since replacement of culture media with media containing EDTA induced a reduction in TEER value of approximately 45% over 90 min (Appendix A). The effect was reversible on further replacement with EDTA-free media which allowed recovery of barrier function.

The formation of tight junctions by HEC-1A cells was investigated by immunostaining of cells cultured on glass slides or in dual chamber inserts. Expression of the tight junction scaffold protein *zona occludens* 1 (ZO-1) was determined using a FITC-conjugated anti-ZO-1 antibody followed by fluorescence microscopy. In cells cultured on glass slides, intense bands of fluorescence surrounding the periphery appeared as a series of apparent fusion points between cells (Figure 1, top panels). Similarly, following immunostaining with anti-ZO-1 antibody, confocal microscopy indicated intense bands of fluorescence at the periphery of cells cultured in Transwell^®^ inserts (Figure 1, lower panel). This figure also shows that HEC-1A cells grow as an irregular multi-layer structure ranging from 1–4 cells in thickness, as reported previously [24].

### 3.2. Permeability of Antiretroviral Drugs across the HEC-1A Model Epithelium

Bi-directional permeability of tenofovir, darunavir and dapivirine across the HEC-1A barrier model was investigated. Based on the findings described above, analyses were performed using HEC-1A layers cultured between 6 and 10 days with requirements that TEER values were >150 Ω × cm^2^ and that deviation from initial TEER values was <20% at the end of the assay. The integrity of the HEC-1A barrier was confirmed by simultaneous measurement of mannitol permeability with P_app_ <4 × 10^−6^ cm/s (flux linearity r^2^ = 1).

Absorptive and secretory permeabilities of tenofovir, determined over a range of concentrations (0.1–100 µM), were P_app_ = 1.76–2.06 × 10^−6^ cm/s and 1.46–1.64 × 10^−6^ cm/s, respectively (Figure 2A). Tenofovir transport was concentration independent and there was no significant difference between absorptive and secretory P_app_ (*p* > 0.05, two-tailed *t*-test) with efflux ratio (secretory/absorptive) of 0.77–0.83. These values were comparable to those determined for mannitol (absorptive and secretory P_app_ = 2.09–3.10 × 10^−6^ cm/s and 2.70–2.85 × 10^−6^ cm/s, respectively) (Figure 2B). Recovery of tenofovir from the apical compartment after absorptive permeability assays was approximately 96–97% of the initial concentration with ≤0.5% associated with HEC-1A cells (Table 2).

Darunavir (logP = 2.82; DrugBank https://go.drugbank.com/drugs/DB01264, accessed on 1 September 2022) is poorly soluble in aqueous buffer and 0.01% DMSO was included for solubilisation in both donor and acceptor compartments. No effects of DMSO on TEER values or mannitol permeability were evident as determined by comparison with control cultures without DMSO (data not shown). Permeabilities of darunavir (concentration range 0.1–20 µM) were again concentration-independent and similar in absorptive and secretory directions although approximately 6-fold greater than those of tenofovir with P_app_ = 9.66–11.20 × 10^−6^ cm/s and 8.59–9.42 × 10^−6^ cm/s, respectively (Figure 2C,D) and efflux ratio of 0.80–0.98. Recovery of darunavir from the apical compartment after absorptive assays was approximately 83–85% with ≤3% cell-associated (Table 2).

These findings indicate that permeation of both tenofovir and darunavir across the HEC-1A model epithelium is transporter-independent. Apparent permeability values for tenofovir were consistent with paracellular diffusion while those for darunavir were indicative of transcellular diffusion.

The permeability of dapivirine was also assessed in this model. Dapivirine is only sparingly soluble in aqueous buffer (logP = 5.6; DrugBank https://go.drugbank.com/drugs/DB08639, accessed on 1 September 2022) and was dissolved in transport buffer supplemented with 1% DMSO. In initial experiments, to investigate bi-directional permeability, the inclusion of 1% DMSO in the donor compartment was associated with increased secretory permeability of mannitol (P_app_ = 5.10–6.81 × 10^−6^ cm/s) but had no effect on absorptive permeability of mannitol (P_app_ = 2.50–2.61 × 10^−6^ cm/s) (Figure 2E). Consistent with this observation, TEER values were reduced by 60–70% when a buffer containing 1% DMSO was applied to the basolateral compartment but there was no reduction when the same buffer was applied to the apical compartment. Subsequently, the permeability of dapivirine was determined in the absorptive direction only. In previous studies of drug permeability using the Caco-2 colorectal cell line model 6, 8, the inclusion of Poloxamer 407 in the receiver compartment increased post-assay recovery of dapivirine. When 0.2% (*w*/*v*) Poloxamer 470 was added in the basolateral receiver compartment, the absorptive permeability of dapivirine increased from P_app_ = 2.50–2.61 × 10^−6^ cm/s (no Poloxamer 470) to 9.52–11.51 × 10^−6^ cm/s × 10^−6^ cm/s (Figure 2F). As shown in Table 2, recovery of dapivirine from the apical compartment was approximately 15–16% (no Poloxamer) and 19–21% (0.2% Poloxamer) with the increased flux reflected in increased drug levels in the receiver compartment and some reduction in cell-associated drug (Table 2).

#### Coadministrations of ARV Drugs

Combinations of ARVs may be more effective than single drugs in the prevention of infection with HIV. To investigate potential drug-drug interactions, the permeability of each labelled drug at a fixed concentration (10 µM) was determined in the presence of either of the other (non-labelled) drugs at the same concentration range as above i.e., 0–10 µM for darunavir and dapivirine and 0–100 µM for tenofovir. No drug-drug interactions were evident. Coadministration with darunavir did not significantly alter the absorptive or secretory permeability of tenofovir and, conversely, tenofovir had no effect on permeabilities of darunavir. Absorptive permeability of tenofovir and darunavir was not altered by dapivirine (secretory permeability of dapivirine was not determined) while absorptive permeability of dapivirine was not affected by tenofovir or darunavir (Appendix A).

### 3.3. Effect of pH on Darunavir Permeability

In view of our previous finding of efflux transporter-dependent permeability of darunavir across the colorectal epithelium and given previous reports of increased drug transporter activity at acidic pH [10,11], we sought to determine the effect of low pH on darunavir transport across our model of the cervicovaginal epithelial barrier. Vaginal pH is acidic in the range 3.5–5.0 [25,26]. As described above, HEC-1A cells were cultured under conditions that allowed adaptation to lower pH by incremental reduction of pH of the medium from 7.4 to 5.5. This was the lowest pH compatible with complete cell viability. The permeability of darunavir was determined in dual chamber cultures with the epithelial barrier formed by low-pH-adapted HEC-1A cells. The pH of the medium in the basal compartment was pH 7.5 for all experiments whereas medium at pH 7.5 or pH 5.5 was added to apical compartments. Non-adapted cell cultures in pH 7.5 medium both in the apical and basal compartments were included as controls. Levels of darunavir in the donor and receptor compartments were determined at a single time-point (2 h) by LC-MS/MS. At pH 7.5 in the absorptive direction, 20 ±2.8% of darunavir transferred to the receiver compartment whereas significantly higher proportions (30.1 ± 6.0%, *p* < 0.001) of darunavir transferred at pH 5.5 (Figure 3). The addition of MK571, a non-specific inhibitor of MRP transporters, to the donor compartment reduced the absorptive transfer of darunavir to the level observed at pH 7.5 (Figure 3). No effects of altering pH in the apical compartment were evident in drug transfer from basolateral to apical compartments. Thus, low pH increases the transfer of darunavir across the epithelial barrier mediated at least in part by increased MRP transporter activity.

### 3.4. Effect of Pro-Inflammatory Cytokines on Drug Permeability

We investigated whether drug permeability was altered in an in vitro model of vaginal inflammation. HEC-1A monolayer cultures in multi-well plated were stimulated with ligands for toll-like receptors (TLRs). From a panel of 8 PAMPs, poly(I:C) induced production of IL-6, IL-8, IL-1α, GM-CSF and TNF-α while flagellin stimulated production of IL-8, GM-CSF and TNF-α (Figure 4A). These findings are consistent with activation induced by TLR3 and TLR5, respectively [27,28]. The optimum concentration for cytokine stimulation by poly(I:C), determined to be 25 µg/mL, was used in subsequent permeability analyses (Appendix A).

HEC-1A cell cultures in dual chambers were then stimulated at the apical surface with poly(I:C) for 18–24 h and cytokine levels were determined in aliquots of culture media from both apical and basolateral compartments. As shown (Figure 4B), concentrations of IL-6 and TNF-α were much higher than those reported previously in both cervicovaginal and cervical fluids during pregnancy while levels of IL-8 were similar to those reported in cervicovaginal fluid [29]. Concentrations of GM-CSF were also comparable to those reported in cervico-vaginal fluid from healthy women [30]. With the exception of IL-8, secretion of cytokines appeared to be polarised to the apical surface.

The effects of poly(I:C) stimulation on the permeability of tenofovir and darunavir were assessed in the same cultures. Apparent permeability values for 10 µM [^14^C]tenofovir and 10 µM [^14^C]darunavir are shown. Figure 4C indicates that there were no significant differences (*p* > 0.05, ANOVA) in P_app_ values determined for apical to basolateral or basolateral to apical directions for tenofovir and darunavir when compared to the permeability across unstimulated HEC-1A cell layers. The P_app_ values for tenofovir were 1.06 ± 0.05 × 10^−6^ cm/s when cells were unstimulated and 1.45 ± 0.21 × 10^−6^ cm/s when cells were stimulated. Additionally, darunavir P_app_ was 8.24 ± 0.60 × 10^−6^ cm/s and 8.45 ± 0.68 × 10^−6^ cm/s, when cells were unstimulated and stimulated. The paracellular tracer molecule mannitol further confirmed that there were no significant (*p* ≥ 0.05, ANOVA) effects of poly(I:C) stimulation on barrier integrity as [^3^H]mannitol P_app_ values were 1.43–2.17 × 10^−6^ cm/s for unstimulated cells and 2.23–2.47 × 10^−6^ cm/s for stimulated cells (Figure 4D).

In a separate model, we investigated whether direct stimulation of VK2/E6E7 cells with pro-inflammatory cytokines (TNF-α, IL-1β or IL-6) altered the permeability of digoxin, an extensively characterised substrate of the P-gp efflux transporter [31]. VK2/E6E7 cells do not form a barrier epithelium and therefore drug permeability was measured by intracellular uptake of digoxin. As shown in Figure 5, intracellular uptake of digoxin was significantly reduced following stimulation with TNF-α or IL-1β but not IL-6, consistent with increased P-gp activity mediated by TNF-α or IL-1β.

### 3.5. Environmental Effects on Vaginal Drug Transporter Expression

We investigated the influence on the expression of a panel of seventeen drug transporters associated with the uptake or efflux of ARV drugs by a range of factors representative of the vaginal environment, namely pro-inflammatory cytokines, acidic pH, microbes associated with health and disease, and female sex hormones.

HEC-1A cells were incubated separately with TNF-α, IL-1β, IL-6 or IL-8 at concentrations corresponding to the maximum levels of these cytokines determined in the cervicovaginal fluid of bacterial vaginosis patients [14,15]. No changes in expression of any transporters were evident in HEC-1A cells (data not shown). In contrast, stimulation of the VK2/E6E7 cell line with TNF-α, IL-1β or IL-6 resulted in increased levels of mRNA encoding the P-gp efflux transporter for which darunavir may be a substrate (Figure 6A–C). TNF-α, IL-1β or IL-6 were also shown to induce the expression of uptake transporters (OATP8 and CNT3) also relevant to darunavir transport (Figure 6A–C).

In cells cultured at pH 5.5, there were no significant changes in the expression of the transporters compared to cells cultured at pH 7.5, other than an increase (2.51 ± 0.67-fold, *p* < 0.05) in the expression of the uptake transporter SLCO4A1 (data not shown).

We investigated the effect of *Lactobacillus crispatus* on drug transporter expression in view of the well-documented role of this bacterial species in vaginal and reproductive health [32]. Co-culture of HEC-1A cells with *L. crispatus* induced minimal changes in drug transporter expression. Of the 17 drug transporters assessed only BCRP showed a change in expression (2.70 ± 0.57-fold, *p* < 0.005) (Table 3). Co-culture of VK2/E6E7 cells with *L. crispatus* resulted in larger effects. Expression of the SLC drug transporters CNT3 (2.32 ± 0.67, *p* < 0.05) and OCT1 (2.44 ± 0.82, *p* < 0.05) increased while expression of OATP8 (3.17 ± 1.20, *p* < 0.05) was induced (Table 3).

We then analysed the effect of *Candida albicans* as a prototypic opportunistic vaginal pathogen. As described above, MoIs of 1 and 0.25 were used for the co-culture of *C. albicans* with HEC-1A and VK2/E6E7 cells, respectively. Downregulation of MRP4 (−2.18 ± 0.22, *p* < 0.001) and ENT2 (−2.12 ± 0.10, *p* < 0.001) was evident in HEC-1A cells (Table 3). In VK2/E6E7 cells, MRP3 was upregulated (2.31 ± 0.83, *p* < 0.05) together with OATPE (5.15 ± 0.27, *p* < 0.001) (Table 3).

In view of the important role of female sex hormone levels on the pharmacokinetics of vaginal ARVs [33], we studied the direct effect of estradiol and progesterone on drug transporter expression. HEC-1A and VK2/E6E7 cells were incubated with β-estradiol or progesterone, respectively, before assessment of mRNA expression of the panel of drug transporters. In HEC-1A cells, stimulation with β-estradiol increased the expression of the efflux transporter BCRP (2.64 ± 0.44, *p* < 0.005) (Figure 7A) in keeping with a previous study in placental cells 16. The same treatment in VK2/E6E7 cells increased expression of MRP3 (2.14 ± 0.30, *p* < 0.005) and CNT3 (3.70 ± 1.43, *p* < 0.05) and downregulated expression of OATPE (2.00 ± 0.25, *p* < 0.005) (Figure 7B). Following stimulation with progesterone, no changes in drug transporter gene expression were detected in HEC-1A cells while in VK2/E6E7 cells, the only change was increased expression of MRP3 (2.54 ± 0.48, *p* < 0.005) (Figure 7C).

## 4. Discussion

In a range of cell lines and tissue explants reflective of sites of HIV-1 mucosal transmission, we have previously demonstrated expression of both efflux and uptake transporters [2,3,34,35] for which reverse transcriptase inhibitors and protease inhibitors may be substrates [2,8,36]. In this study, we sought to investigate the permeability and mode of transport of three candidate ARV drugs for topical pre-exposure prophylaxis in an in vitro model of the cervicovaginal epithelial barrier. Notwithstanding the site-specific variation of expression of drug transporters along the cervicovaginal tract [2], we elected to use the endometrial cell line HEC-1A in view of the known ability of this cell line to form tight junctions [37,38]. The formation of tight junctions in this study was confirmed by immunofluorescence. Confocal microscopy also indicated that HEC-1A cells in culture formed an irregular multilayer, possibly more reflective of the multi-layered epithelium of the vaginal and ectocervix surfaces of the female genital tract. The abrupt decrease in transepithelial electrical resistance following the removal of calcium from HEC-1A cultures as well as the recovery after calcium replacement provided further evidence for the formation of tight junctions. Transepithelial electrical resistance was higher in cells grown on polyester compared with polycarbonate membranes, suggesting that polyester is best suited for the preservation of tight junctions. The finding that pre-coating with either laminin or Matrigel had little effect on barrier formation in HEC-1A cultures has also been reported in other models [39] and may reflect the intrinsic production of extracellular matrix proteins [40]. Direct culture of HEC-1A cells on polyester membranes with no addition of such proteins may provide a more standardised system to measure drug transport.

We applied the HEC-1A model to investigate the permeability of tenofovir, darunavir and dapivirine. For the studies reported here, permeability was determined in both absorptive and secretory directions as significant differences in apparent permeability values for either direction can be presumed to reflect the effects of drug transporters. In contrast, previous studies have reported tenovofir permeability in the absorptive direction only [9,41]. In this study, all three ARVs showed transporter-independent permeability across the epithelium. This included darunavir in contrast with previously reported findings in the Caco-2 cell model of colorectal epithelial barrier that darunavir uptake is P-gp-dependent [8]. Apparent permeability values and drug intracellular accumulation for each drug reflected the physiochemical property of each drug. Absorptive permeability data of tenofovir, a negatively charged hydrophilic molecule, suggests that this drug slowly permeates *via* the paracellular route through pores of the tight junctions with negligible intracellular accumulation. Darunavir is lipophilic but our data suggests that it efficiently permeates *via* transcellular passive diffusion with low intracellular accumulation. Despite the low solubility of darunavir in aqueous buffers, this drug did not show the properties of a highly lipophilic compound as evident from its low intracellular HEC-1A accumulation. The data here suggests that darunavir freely permeates through HEC-1A cell layers *via* passive diffusion without binding to lipids within cell membranes. In contrast, dapivirine was confirmed as a highly hydrophobic and lipophilic molecule that permeates through transcellular passive diffusion with high intracellular accumulation. Although dapivirine shared a similar basolateral accumulation value to darunavir (in the presence of poloxamer 407), the much higher intracellular accumulation reflects the difference in lipophilicity between the two drugs. The low absorptive P_app_ of dapivirine is consistent with partition and accumulation of uncharged drugs at high levels in the membrane compartment of epithelial cells. Such accumulation has also been shown in vaginal epithelial tissue following vaginal administration in rabbit and macaque models [42].

No drug-drug interactions were observed when ARV drugs were co-administered in double combinations. The apparent permeability values for tenofovir in both directions determined in this study are in good agreement with those reported previously for tenovofir permeability through HEC-1A dual chamber cultures [9]. Findings were also in agreement with a study of tenofovir permeability through excised ectocervical tissue following exposure to tenofovir gel preparations, notwithstanding the inter- and intra-patient variability in diffusion profiles [41]. In our previous study [2], we reported a good association between HEC-1A cells and human cervical explants in terms of expression of efflux transporters for which ARV drugs are substrates. On the other hand, expression of SLC transporters in HEC-1A was only in part consistent with expression in tissue explants. Notably CNT3, potentially relevant for transport of tenofovir [43], was expressed in human tissue explants but not HEC-1A cells. On this basis, the application of the HEC-1A model to the study of ARV drugs which are substrates for CNT3, for example nucleoside inhibitors, should be interpreted in the context of a discrepancy in the expression of this uptake transporter.

We sought to investigate ARV drug permeability and transporter-dependent activity in conditions simulating the cervicovaginal environment. Indeed, previous studies have reported an increase in the functional activity of certain drug transporters in other cell types exposed to acidic pH [10,44]. Cell culture medium acidification in the absence of drugs had no effect on the expression of genes encoding drug transporters. However, in the dual chamber model of drug permeability, medium acidification to pH 5.5 in the apical compartment, to reflect the physiological vaginal pH as closely as possible, increased absorptive transfer of darunavir, suggesting that Pg-p dependent darunavir transport is only active in acidic culture conditions. Further, inhibition of MRP efflux transporters (for which darunavir is a substrate) *via* MK571 reduced absorptive transfer of darunavir to the level observed at neutral pH suggesting that low pH increases the transfer of darunavir across the epithelial barrier, mediated at least in part by increased MRP transporter activity. This finding is supported by our previous study showing increased MRP2 expression in a macaque model of intravaginal stimulation with film-released darunavir [3] and previous data showing darunavir as a substrate for MRP transporters in other cell types [36]. On the other hand, the increased MRP activity may be also driven by intracellular events, such as a decrease in intracellular calcium concentrations, which in turn inhibit protein kinase C activity as proposed for P-gp [10]. The effect of acidic pH on the permeability of tenofovir and dapivirine was not evaluated in this study.

Topical microbicides may be administered under conditions of vaginal inflammation that result from bacterial vaginosis or sexually transmitted diseases. Pro-inflammatory cytokines have been shown to increase epithelial permeability in a variety of systems by reducing tight junction activity [45]. Mechanisms of tight junction inhibition mediated by inflammation include downregulation and relocalisation of tight junction proteins by TNF-α and IL-6, respectively [46,47]. Here the potential for modelling inflammation and the study of the effect of inflammation on drug permeability was studied by stimulation of HEC-1A cells with ligands for TLR-3 and TLR-5. In HEC-1A cell layers stimulated with the TLR ligand poly(I:C), levels of pro-inflammatory cytokines were comparable to those previously reported to reduce maintenance of tight junctions, however no increase in permeability was evident. Inflammation did not affect ARV drug permeability across the epithelial barrier and neither it did affect the expression of drug transporters in HEC-1A cells. In contrast, direct stimulation of vaginal cell line VK2/E6E7 with pro-inflammatory cytokines TNF-α or IL-1β reduced intracellular uptake of P-gp substrate digoxin pointing to increased activity of this efflux transporter. Further, stimulation of VK2/E6E7 cells with TNF-α, IL-1β or IL-6 resulted in increased levels of mRNA encoding P-gp. This finding was consistent with the effect observed in peripheral mononuclear blood cells [16]. Increased P-gp activity in vaginal cells under pro-inflammatory conditions may result in reduced absorption of ARV drugs for which P-gp is a substrate. Nonetheless, the lack of any effect on ARV drug permeability and drug transporters expression in HEC-1A cells, suggests that the effect of inflammation on drug transporter activity and expression may be dependent on the cell type in the cervicovaginal tract. Indeed, patterns of drug transporter expression vary along the cervicovaginal tract [2] and the endometrial epithelium-derived HEC-1A cell line may not reflect the pattern of expression of some transporters in vaginal or cervical epithelia.

We identified vaginal microbial species as well as female sex hormones as potentially important factors for the modulation of drug transporter expression. As noted for cell stimulation with pro-inflammatory cytokines, the effect on drug transporters was more pronounced in VK2/E6E7 cells compared to HEC-1A cells in both experiments of cell stimulation with microbes and sex hormones. Overall, VK2/E6E7 stimulation with either *L. crispatus* or *C. albicans* resulted in a more evident induction of expression of SLC uptake transporters compared to ABC efflux transporters. Increased expression of uptake transporters could increase absorption of ARVs for which the uptake transporters are substrates, thus dampening any increased efflux transporter activity induced by low pH, inflammation and hormonal changes.

## 5. Conclusions

Taken together the findings of this study demonstrate the comprehensive application of a robust and physiologically relevant in vitro model for the testing of optimized formulations of ARV-based vaginal microbicides by allowing rapid measurement of drug transporter dependency, drug-drug interactions and the effect of environmental factors on tissue distribution. The HEC-1A model was validated for the study of ARV drug permeability in acidic conditions reflective of the cervicovaginal environment. On the other hand, the study of the effect of inflammation, and possibly vaginal microbes and female sex hormones, on the expression and activity of drug transporters should be more appropriately conducted in vaginal cells.

## Figures and Tables

**Figure 1 pharmaceutics-14-01938-f001:**
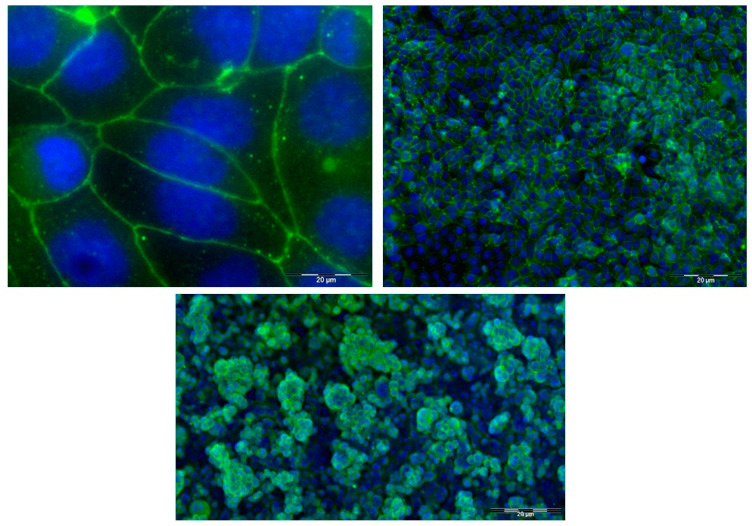
Immunofluorescence staining of ZO-1 expression in HEC-1A cells. ZO-1 expression was visualised using a mouse IgG1 FITC-conjugated monoclonal antibody to the ZO-1 tight junction protein (green), and cell nuclei were stained with Dapi (blue). Merge images were produced of cell nuclei and tight junction staining at 10× and 60× magnifications for cells grown on glass slides for five days (**top panels**) and HEC-1A cell layers cultured for 7 days on polyester Transwells^®^ (**lower panel**). Scale bar for all images in 20 µm.

**Figure 2 pharmaceutics-14-01938-f002:**
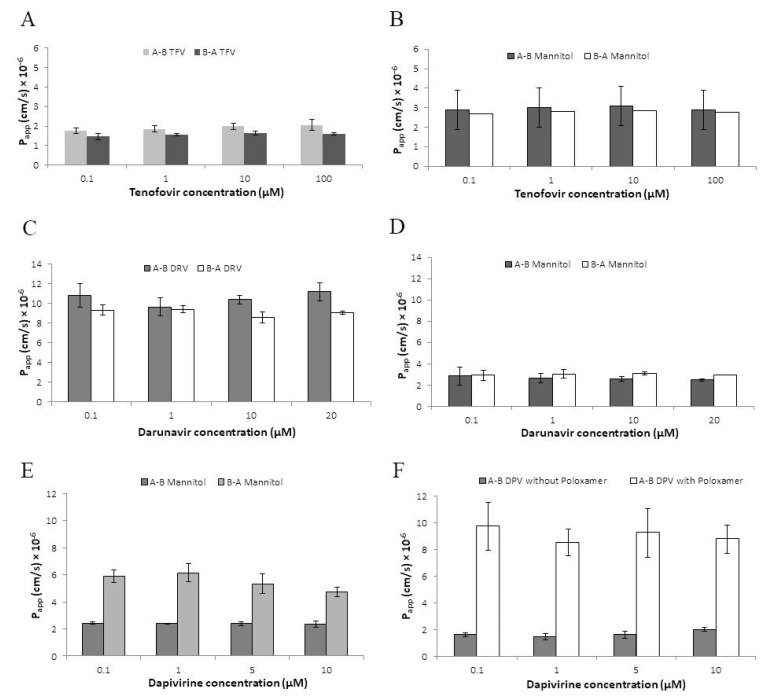
Permeability of tenofovir, darunavir and dapivirine in the HEC-1A model. (**A**) Absorptive (A-B TFV) and secretory (B-A TFV) permeabilities of tenofovir presented as P_app_ values and shown over a range of concentrations of tenofovir (0.1–100 µM). (**B**) Absorptive and secretory permeabilities of mannitol used as paracellular transport marker in the tenofovir permeability assay. (**C**) Absorptive (A-B DRV) and secretory (B-A DRV) permeabilities of darunavir presented as P_app_ values and shown over a range of concentrations (0.1–20 µM). (**D**) Absorptive and secretory permeabilities of mannitol used as paracellular transport marker in the darunavir permeability assay. (**E**) Absorptive (A-B) and secretory (B-A) permeabilities of mannitol used as paracellular transport markers in the initial dapivirine permeability assay. P_app_ values are shown over a range of concentrations of dapivirine (0.1–10 µM) solubilised in 1% DMSO. (**F**) Effect of Poloxamer 470 on absorptive permeabilities of dapivirine. Absorptive permeabilities of dapivirine (A-B DPV) over the range of dapivirine concentrations are shown in the presence and absence of Poloxamer 470 added to the received compartment. The standard deviation of triplicate measurements is shown.

**Figure 3 pharmaceutics-14-01938-f003:**
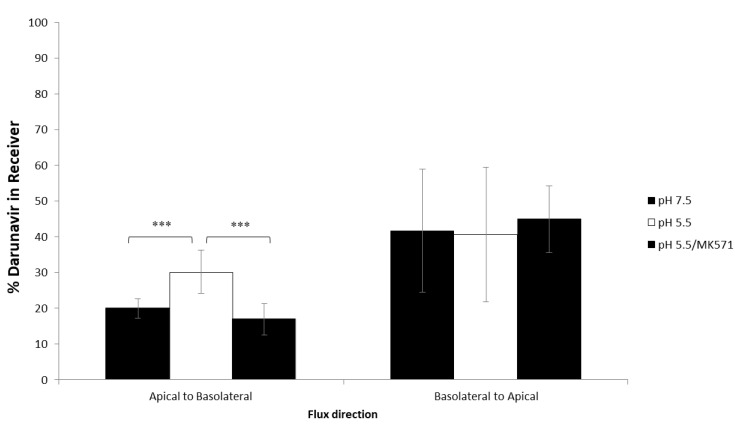
Transcellular permeability of darunavir across an HEC-1A monolayer cultured at pH 7.5 or pH 5.5. The apical to basolateral and basolateral to apical permeability of darunavir across HEC-1A monolayers cultured at pH 7.5 or 5.5 in the apical compartment is shown. The apical to basolateral and basolateral to apical permeability of darunavir in the presence of the MK571 inhibitor across HEC-1A monolayers cultured at pH 5.5 is shown in comparison with permeability at pH 7.5 or 5.5 without inhibitor. Data is presented as % applied darunavir permeated into the receiver chamber as the mean ± SD (*n* = 10). Statistical analysis performed was a one-way ANOVA using Bonferroni post-hoc test where; *** *p* < 0.001.

**Figure 4 pharmaceutics-14-01938-f004:**
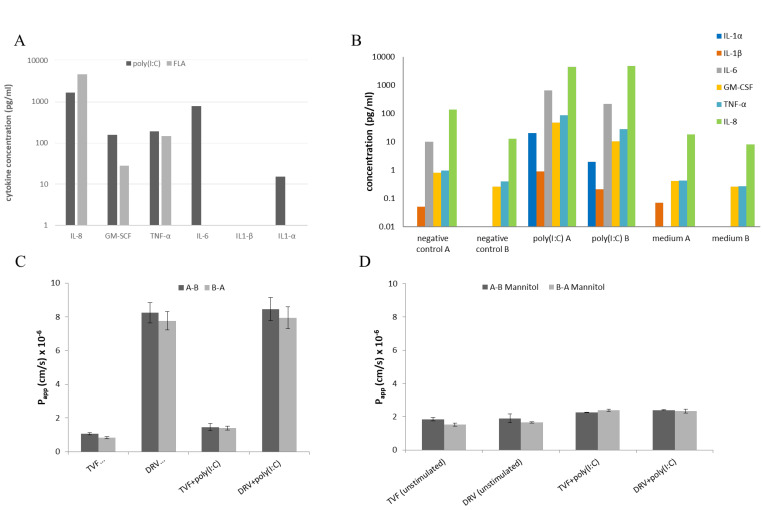
Effect of PAMP stimulation on pro-inflammatory cytokine production and drug permeability in the HEC-1A model. (**A**) Pro-inflammatory cytokine production in response to stimulation with poly(I:C) and flagellin (FLA). (**B**) Pro-inflammatory cytokine production in the donor (**A**) and receiver (**A**) compartments on stimulation with poly(I:C) in comparison with stimulation with an irrelevant protein and medium alone. (**C**) Absorptive (A-B) and secretory (B-A) permeabilities of tenovofir (TNF) and darunavir (DVR) in cells stimulated with poly(I:C) (25 µg/mL) in comparison with unstimulated cells. (**D**) Absorptive and secretory permeabilities of mannitol used as paracellular transport markers in the tenofovir and darunavir permeability assays. Measurements were carried out in triplicate.

**Figure 5 pharmaceutics-14-01938-f005:**
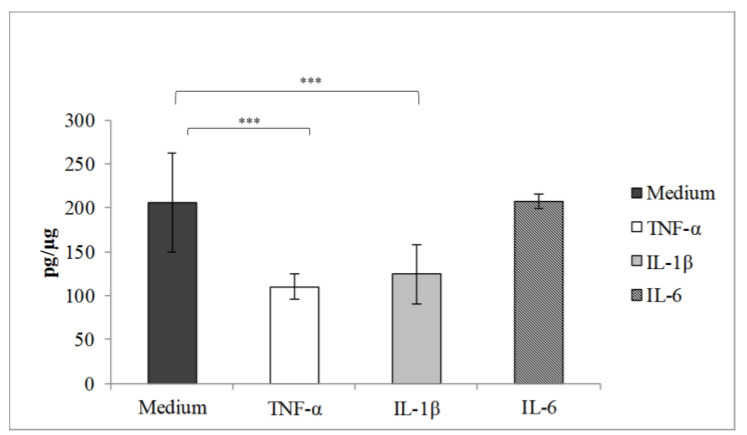
Intracellular accumulation of digoxin in VK2/E6E7 cells stimulated with cytokines. The accumulation of digoxin in VK2/E6E7 cells post-cytokine stimulation is shown in comparison to the unstimulated control. The data is presented as digoxin accumulated (pg/µg) ± SD (*n* = 6). Statistical analysis was a one-way ANOVA using Bonferroni correction, *** *p* < 0.001.

**Figure 6 pharmaceutics-14-01938-f006:**
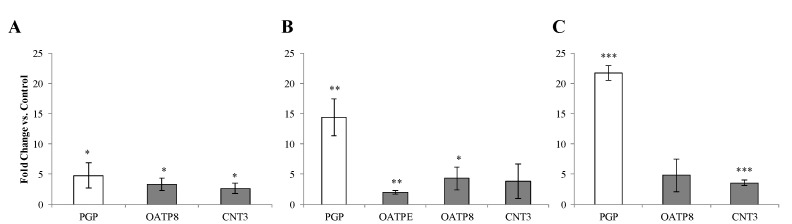
Drug transporter mRNA expression in VK2/E6E7 cells stimulated with pro-inflammatory cytokines. Quantitative PCR of cDNA was carried out using the SYBRgreen^®^ chemistry. (**A**) TNF-α (**B**) IL-1β (**C**) IL-6 for 72 h. White bars represent ABC transporters. Grey bars represent SLC transporters. Data is presented as fold change versus medium control ±SD (*n* = 3). Two-tailed students *t*-test * *p* < 0.05, ** *p* < 0.005, *** *p* < 0.001.

**Figure 7 pharmaceutics-14-01938-f007:**
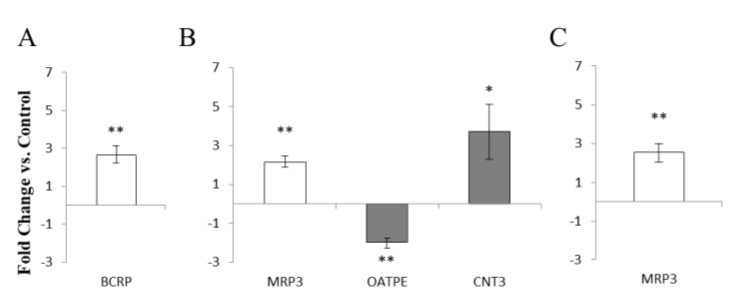
Modulation of drug transporter mRNA expression in cells stimulated with female sex hormones. (**A**) HEC-1A cells stimulated with 100 nM β-estradiol (**B**) VK2/E6E7 cells stimulated with 100 nM β-estradiol (**C**) VK2/E6E7 cells stimulated with 750 nM progesterone for 72 h. The SYBRgreen^®^ chemistry was used. White bars represent ABC transporters. Grey bars represent SLC transporters. Data is presented as fold-change versus 0.01% ethanol control ± SD (*n* = 3). Statistical analysis was a two-tailed students *t*-test * *p* < 0.05, ** *p* < 0.005.

**Table 1 pharmaceutics-14-01938-t001:** Primer sequences used in gene expression assays.

Drug Transporter	NCBI Ref Sequence	Forward Primer 5′ > 3′	Reverse Primer 5′ > 3′
HPRT1	NG_012329.2	TGACCTTGATTTATTTTGCATACC	CGAGCAAGACGTTCAGTCCT
RPLP0	NC_000012.12	TCCAGGCTTTAGGTATCACCAC	TTGATCAGCTGCACATCACTC
P-gp	NG_011513.1	CTTCATCGAGTCACTGCCTAATAA	GCTATGGCAATGCGTTGTT
BCRP	NG_032067.2	TGGCTTAGACTCAAGCACAGC	TCGTCCCTGCTTAGACATCC
MRP1	NG_028268.2	CCTGTTCAACGTCATTGGTG	AGGAAGCCACGTAGAACCTCT
MRP2	NG_011798.2	CTGGATCACCTCCAACAGGT	AGAAGACAGTCAGGTTCCCAAC
MRP3	NC_000017.11	AATGTCGACCCTAACCCCTAC	GGGTCAGGGTTTAGCAGCTT
MRP4	NG_050651.2	GCGCCTGGAATCTACAACTC	CTTTGTATGCCCGGATGGT
MRP5	NG_047115.1	CAGCGACCTGACGGAGAT	GGGCAAGGCTGATCCTCT
MRP6	NG_007558.3	CGGGAAGGATTGCATCAC	GGCACCGTGAGGTTTATTCT
MRP7	NC_003074.8	GCCCTCAATGATGACCTCAG	CCTCCGCTAAGGGTGACA
ENT1	NG_042893.1	TCACCAGCCTCAGGACAGAT	GAGTGGCCGTCATGAAAAA
ENT2	NC_000011.10	GCTGAGCTCCTCCAGTCTGA	GGCTCATCTGGCTCTGATTC
CNT3	NC_000009.12	CTGAACTCCACGCCATCA	CAAGTGGGAGGATGGAACC
OCT1	NC_000006.12	CTCGCCCAACTACATGTCC	CGAGCCAACAAATTCTGTGAT
OCT3	NC_000006.12	CACCATCGTCAGCGAGTTT	CAGGATGGCTTGGGTGAG
OATPD	NC_000015.10	TGTTCCCTATGGAAACAGCA	CAGGTAGGTGATGCCATCTG
OATPE	NC_000020.11	AGCTGCCACCTTGTTTGG	GCCTGAGCTTGTTCACAAAGA
OATP8	NG_032071.1	TCAAGTGGTATTAAAAAGCATACAGTG	TTCACCCAAGTGTGCTGAGT

**Table 2 pharmaceutics-14-01938-t002:** Cell-associated drug and distribution between the apical and basolateral compartments on evaluation of absorptive permeability.

Drug Name (Concentration)	% Drug Distribution *		
	Apical	Basolateral	Cell-Associated
TFV (0.1 μM)	97.03 ± 0.28	2.63 ± 0.25	0.34 ± 0.06
TFV (1 μM)	96.85 ± 0.29	2.73 ± 0.24	0.42 ± 0.08
TFV (10 μM)	96.91 ± 0.27	2.79 ± 0.25	0.3 ± 0.06
TFV (100 μM)	96.59 ± 0.41	2.99 ± 0.36	0.42 ± 0.17
DRV (1 μM)	82.90 ± 1.76	15.13 ± 1.44	1.97 ± 1.97
DRV (5 μM)	83.15 ± 2.61	14.55 ± 1.88	2.55 ± 1.22
DRV (10 μM)	83.36 ± 2.12	13.95 ± 1.75	2.70 ± 1.31
DRV (20 μM)	81.72 ± 1.61	15.31 ± 1.22	2.97 ± 1.27
DPV (0.1 μM)	15.04 ± 0.55	2.87 ± 0.26	82.10 ± 0.30
DPV (1 μM)	14.72 ± 0.10	2.13 ± 0.33	83.15 ± 0.35
DPV (5 μM)	14.98 ± 0.76	1.86 ± 0.41	83.16 ± 1.15
DPV (10 μM)	14.52 ± 0.45	2.09 ± 0.18	83.39 ± 0.55
DPV (0.1 μM) + Poloxamer	16.06 ± 3.26	10.56 ± 1.26	73.37 ± 3.37
DPV (1 μM) + Poloxamer	16.47 ± 2.86	11.01 ± 1.41	72.51 ± 2.86
DPV (5 μM) + Poloxamer	17.47± 2.76	10.78 ± 1.18	71.75 ± 2.33
DPV (10 μM) + Poloxamer	17.59 ± 3.41	11.72 ± 1.34	70.69 ± 2.76

* The standard deviation of triplicate measurements is shown. TFV = tenofovir, DRV = darunavir, DPV = dapivirine.

**Table 3 pharmaceutics-14-01938-t003:** A summary of changes in drug transporter mRNA expression in HEC-1A cell and VK2/E6E7 cells upon 24 h co-culture with *Lactobacillus crispatus* or *Candida albicans*.

	*Lactobacillus crispatus*		*Candida albicans*	
	HEC-1A	VK2/E6E7	HEC-1A	VK2/E6E7
P-GP	-	-	-	-
BCRP	↑	-	-	-
MRP1	-	-	-	-
MRP2	-	-	-	-
MRP3	-	-	-	↑
MRP4	-	-	↓	-
MRP5	-	-	-	-
MRP6	-	-	-	-
MRP7	-	-	-	-
OATP8	-	↑↑	-	-
OATPD	-	-	-	-
OATPE	-	↑↑	-	↑↑↑
OCT1	-	↑	-	-
OCT3	-	-	-	-
CNT3	-	↑	↑	-
ENT1	-	-	-	-
ENT2	-	-	↓	-

-, no change; ↑/↓ fold change 2–3; ↑↑/↓↓ fold change 3–5; ↑↑↑/↓↓↓ fold change > 5; Positivity threshold was Ct < 35. Measurements were carried out in triplicate.

## Data Availability

All original output data can be made available on request.

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
