# Peer review of "Comprehensive Study of Antiretroviral Drug Permeability at the Cervicovaginal Mucosa via an In Vitro Model"

_pharmaceutics, 2022, doi:10.3390/pharmaceutics14091938_

Round 1
Reviewer 1 Report
Constandinos Carserides et al investigated the in vitro cervicovaginal model to determine the antiretroviral drug permeability. The authors evaluated the TEER measurements, drug permeability, cell stimulation with drugs, cytokines, hormones, microbes, and other studies in this manuscript. These results are of great interest for the environmental effects on drug permeability and transporter expression levels, especially in the HIV field.
Below are the comments and include the description in the manuscript.
1. Citation format should be modified. The citations are not proper in the manuscript and are mixed with the sentences.
2. In the 2.1 section, Purity information of the reference standards and drugs should be included. Manufacturer and location information are needed for several reagents, chemicals, and strains.
3. Check the spelling mistakes, and superscripts in the manuscript and correct them.
4. Gene accession numbers should be included in table 1. Why didn’t the authors use GAPDH as a housekeeping gene?
5. Delete line 314.
6. How the current in vitro cell permeability model is physiologically biorelevant compared to in vivo human vaginal absorption and ex vivo human tissue permeability? For physiological relevance, How the permeability results of this in vitro model can be compared/correlated to EPIvaginal models and ex vivo human vaginal tissue explant models?
7. How are transporters' expression levels in this model correlated to the human vaginal tissues?
8. The authors should mention the number of replicates used for all the experimental results in the figure and table legends.
9. Darunavir is a substrate of Pgp transporter and tenofovir is a substrate of MRP4. The efflux ratios of darunavir and Tenofovir using this in vitro model are <1. What would be the reason for this? Did the authors evaluate the activity of uptake and efflux transporters in this model? The authors should discuss this in the manuscript.
10. The authors did not show transporter assays' substrate and inhibitor activity for validating this model.
Reviewer 2 Report
The manuscript submitted by Constandinos Carserides and colleagues investigates effects of pH and inflammation on ARV uptake and drug transporter-dependent permeability across the vaginal barrier epithelium via their developed in vitro model. The current manuscript includes characterization data for the in vitro model and ex vivo data to study the factors that can affect the uptake of topically applied ARVs. The strategy and rationale are good, and the study is worthy to be published in Pharmaceutics. Below comments aimed at improving the manuscript.
1. Minor revision:
1) Line 41 to 45, citation needed.
2) Line 314, is this sentence mistakenly placed here?
3) Figure 1, the scale bar is not clear to the readers. scale bar is too small to see, please also clarify in the figure legend.
4) Line 382, reference error.
5) Please combine Figure 2 and 3. Alternatively, the author could move Figure 2 to the supplementary section since no significant differences have been investigated.
6) Figure 3, please mark any significant differences on the chart.
7) Line 459, duplicate MS.
2. Major revision:
1) In the reviewer’s opinion, the focus of the whole manuscript is not the model, but the factors that can affect the uptake of topically applied ARVs. Then the title would be inappropriate. Please revise, an example could be a “comprehensive study of antiretroviral drug permeability at the cervicovaginal mucosa via an in vitro model”.
2) I cannot see the problem in the field by reading the current introduction. Could the author clearly state the problem? Why do we need a new in vitro model? Is there any other study on this topic? What’s the innovation point of this study? The current version is too weak for me. The author should at least include the importance and novelty of further investigation of the mechanism.
3) Discussion is too surface level. The author can definitely discuss:
a. Why does polyester give higher TEER than polycarbonate membranes?
b. Why permeation of both tenofovir and darunavir across the HEC-1A model is different from dapivirine? Is that because of any special chemical structure differences? Can we use the conclusion of the current study to predict the permeation profile of any other kind of ARVs?
c. Can the author discuss more why low pH may increase the transfer of darunavir across the epithelial barrier? Is that because of the pKa of darunavir?
d. Similarly, for the environmental effect section, a deeper discussion would highly improve the importance of this manuscript.
Round 2
Reviewer 1 Report
The authors significantly improved the manuscript and provided reasonable responses to all the comments.